# Evaluating the screening value of serum light chain ratio, β₂ microglobulin, lactic dehydrogenase and immunoglobulin in patients with multiple myeloma using ROC curves

**Limei Ying[1], Xiaochang Zhang[1], Nina Lu[1], Lei Zhao[2], Yanfang Nie[3], Guofen Wang[4], Sai Chen[1], Linglong Xu👁[1]** *

**1** Department of Hematology, Taizhou Central Hospital (Taizhou University Hospital), Taizhou, Zhejiang, China, **2** Department of Laboratory Medicine, Taizhou Central Hospital (Taizhou University Hospital), Taizhou, Zhejiang, China, **3** Department of Nephrology, Taizhou Central Hospital (Taizhou University Hospital), Taizhou, Zhejiang, China, **4** Department of Reumatology & Immunology, Taizhou Central Hospital (Taizhou University Hospital), Taizhou, Zhejiang, China

* xull@tzc.edu.cn

**Data Availability Statement:** Data cannot be shared publicly because of our hospital's privacy policy. Data are available from the Taizhou Central

## Abstract

### Objective

Several laboratory and imaging assays are required to diagnose multiple myeloma (MM). Serum and urine immunofixation electrophoresis are two key assays to diagnose MM, while they have not been extensively utilized in Chinese hospitals. Serum light chain (sLC), β₂ microglobulin (β₂-MG), lactic dehydrogenase (LDH), and immunoglobulin (Ig) are routinely measured in the majority of Chinese hospitals. Imbalance of sLC ratio (involved light chain/ uninvolved light chain) is frequently observed in MM patients. This study aimed to evaluate the screening value of sLC ratio, β₂-MG, LDH, and Ig in MM patients using receiver operating characteristic (ROC) curves.

### Methods

Data of 303 suspected MM patients, who were admitted to the Taizhou Central Hospital between March 2015 and July 2021, were retrospectively analyzed. In total, 69 patients (MM arm) met the International Myeloma Working Group (IMWG) updated criteria for the diagnosis of MM, while 234 patients were non-MM (non-MM arm). All patients' sLC, β₂-MG, LDH, and Ig were measured using commercially available kits according to the manufacturer's instructions. The ROC curve analysis was employed to assess the screening value of sLC ratio, β₂-MG, LDH, creatinine (Cr) and Ig. The statistical analysis was carried out by SPSS 26.0 (IBM, Armonk, NY, USA) and MedCalc 19.0.4 (Ostend, Belgium) software.

### Results

There was no significant difference between the MM and non-MM arms in terms of gender, age and Cr. The median sLC ratio in the MM arm was 11.5333, which was significantly

Hospital Institutional Data Access / Ethics Committee (contact via zxyykjc@tzzxyy.com) for researchers who meet the criteria for access to confidential data.

**Funding:** Linglong Xu received a grant from Taizhou Science and Technology Bureau (No. 20ywa31) The funders had no role in study design, data collection and analysis, decision to publish, or preparation of the manuscript.

**Competing interests:** The authors have declared that no competing interests exist.

**Abbreviations:** AUC, area under curve; Cr, creatinine; FLC, free light chain; I/U, involved/ uninvolved; IFE, immunofixation electrophoresis; Ig, immunoglobulin; IMWG, International Myeloma Working Group; LDH, lactic dehydrogenase; MM, multiple myeloma; ROC, Receiver operating characteristic; sLC, Serum light chain; β$_2$-MG, β$_2$ microglobulin.

higher than that of 1.9293 in the non-MM arm ($P$<0.001). The area under the curve (AUC) of sLC ratio was 0.875, which indicated a robust screening value. The optimal sensitivity and specificity were 81.16% and 94.87% respectively, when the sLC ratio was set as 3.2121. The serum levels of β$_2$-MG and Ig were higher in the MM arm than those in the non-MM arm ($P$<0.001). The AUC values of β$_2$-MG, LDH, and Ig were 0.843 ($P$<0.001), 0.547 ($P$ = 0.2627), and 0.723 ($P$<0.001), respectively. The optimal cutoff values of β$_2$-MG, LDH, and Ig were 1.95 mg/L, 220 U/L, and 46.4 g/L respectively, in the context of screening value. The triple combination of sLC ratio (3.2121), β$_2$-MG (1.95 mg/L), and Ig (46.4 g/L) yielded a higher screening value compared with that of sLC ratio alone (AUC, 0.952; $P$<0.0001). The triple combination had a sensitivity of 94.20% and a specificity of 86.75%. The addition of LDH to the triple combination and formation of quadruple combination did not optimize the screening value, with AUC, sensitivity, and specificity of 0.952, 94.20%, and 85.47%, respectively.

## Conclusion

The triple combination strategy (sLC ratio, 3.2121; β$_2$-MG, 1.95 mg/L; Ig, 46.4 g/L) is accompanied by remarkable sensitivity and specificity for screening MM in Chinese hospitals.

## Introduction

Multiple myeloma (MM), with a hallmark of proliferation of monoclonal plasma cells, is one of the common hematological malignancies in China [1, 2], especially in the elderly population. With emerging new agents and cell therapy, the survival of MM patients has noticeably prolonged in recent decades. However, the majority of patients with MM experience relapse and could not be treated. Patients with MM may not have overt symptoms specific to MM at very early-stage of the disease, which could compromise the early diagnosis and treatment of the disease. Thus, it is crucial to diagnose MM patients as early as possible. The criteria for MM involve several laboratory and imaging assays, of which immunofixation electrophoresis (IFE) is the key to detect the monoclonal protein as an immunoglobulin (IgG) antibody with a lambda light chain. To date, IFE has not been broadly employed in all hospitals in China, especially in county hospitals and district hospitals, causing a challenge in the early-stage diagnosis of MM. Serum light chain (sLC), β$_2$ microglobulin (β$_2$-MG), lactic dehydrogenase (LDH), and Ig are more routinely utilized than IFE. Remarkable imbalance of sLC ratio (involved light chain/uninvolved light chain) has been frequently reported in MM patients. To date, few studies have concentrated on the screening value of sLC ratio, β$_2$-MG level, LDH level, and Ig level in MM patients. Thus, the present study aimed to evaluate the screening value of these parameters using the receiver operating characteristic (ROC) curves.

## Methods

### Patients

Data of 303 patients with suspected MM who were admitted to the Taizhou Central Hospital (Taizhou, China) between March 2015 and July 2021 were retrospectively analyzed. All patients had abnormal serum Ig or sLC levels. Besides, 69 patients (MM arm) met the International Myeloma Working Group (IMWG) updated criteria for the diagnosis of MM,

while 234 patients were non-MM (non-MM arm) [3]. In addition, the levels of sLC, $\beta_2$-MG, LDH, Cr, and Ig of all patients were recorded. The study was approved by the Ethics Committee of the Taizhou Central Hospital, and it was conducted according to the principle of Declaration of Helsinki. Written informed consents were provided by all participates in the study.

## Measurement of sLC, $\beta_2$-MG, LDH, Cr, and Ig

Briefly, 5~8 mL of peripheral blood was taken from each patient under fasting conditions. Then, the levels of sLC, $\beta_2$-MG, LDH, Cr, and Ig were measured using commercially available kits according to the manufacturer's instructions. The levels of sLC and Ig were measured by the IMMAGE 800 protein chemistry analyzer (Beckman Coulter Inc., Brea, CA, USA) with the accessory light chain kit. Serum levels of total protein, albumin, $\beta_2$-MG, and LDH were tested by the ADVIA 2400 Clinical Chemistry system (Siemens, Munich, Germany) using the total protein assay kit (Medicalsystem Biotechnology Co., Ltd., Ningbo, China), albumin assay kit (Medicalsystem Biotechnology Co., Ltd.), $\beta_2$-MG assay kit (Medicalsystem Biotechnology Co., Ltd.), and LDH assay kit (Medicalsystem Biotechnology Co., Ltd.), respectively. The Ig level was calculated as total protein level minus albumin level. The level of Cr was measured by the LABOSPECT 008AS Clinical Chemistry system (Hitachi, Japan), using the creatinine liquicolor kit (Medicalsystem Biotechnology Co., Ltd., Ningbo, China).

## Statistical analysis

In the present study, SPSS 26.0 (IBM, Armonk, NY, USA) and MedCalc 19.0.4 (Ostend, Belgium) software were used to perform the statistical analysis. Measurement data were expressed as the mean ± standard deviation (for normally distributed data) or median (25-75th percentiles, for abnormally distributed data). Categorical data were expressed as number (percentage). Differences in age, gender, and laboratory parameters were analyzed by the t-test, Chi-square test, and Mann-Whitney U test. The area under receiver operating characteristic (ROC) curve (AUC) was calculated to evaluate the screening value of sLC, $\beta_2$-MG, LDH, and Ig. As for triple combination and quadruple combination, logistic regression analysis was employed to calculate the positive diagnostic rate, followed by calculation of the AUC, sensitivity, and specificity of each variable. ROC curves were compared by the Z test. Statistical significance was set as $P < 0.05$.

## Results

### Participants' baseline characteristics

Among the 303 suspected MM patients, 69 were diagnosed with MM, and 234 were non-MM, of whom, there were 44 men and 25 women in the MM arm, with a median age of 66 (36–86) years old, and a median level of Cr 85.0μmol/L (64.5–110 μmol/L). There were 118 men and 116 women in the non-MM arm, with a median age of 65.5 (24–94) years old, and a median level of Cr 73.5μmol/L (56.8–108.5μmol/L). There were no significant differences in age, Cr and gender between the two arms ($P$>0.05) (Table 1).

### sLC ratio in the two arms

The median sLC ratio in the MM arm was 11.5333, the minimum value was 0.86, and the maximum value was 326.19; the median sLC ratio in the non-MM arm was 1.9293, the minimum value was 0.59, and the maximum value was 17.16. Comparison analysis showed that the sLC

**Table 1. Comparison of the gender, age, Cr, sLC(I/U), $\beta_2$-MG, LDH, Ig levels between the two arms.**

|  | MM arm | Non-MM arm | *P* |
|---|---|---|---|
| **Number** | 69 | 234 |  |
| **Male: Female** | 44:25 | 118:116 | 0.051 |
| **Age** | 66(36–86) | 67(24–94) | 0.114 |
| **Cr(μmol/L)** | 85.0(64.5–110) | 73.5(56.8–108.5) | 0.077 |
| **eGFR(ml/min/1.73m²)** | 73.6(47.5–93.2) | 83.4(49.9–101.1) | 0.181 |
| **sLC(I/U)** | 11.5333(4.0721–54.8929) | 1.9293(1.6027–2.2534) | <0.001* |
| **$\beta_2$-MG(mg/L)** | 4.57(2.95–8.30) | 1.57(1.28–2.36) | <0.001* |
| **LDH(U/L)** | 199(166–262) | 191(165–237) | 0.24 |
| **Ig(g/L)** | 48.7(31.0–72.2) | 32.1(27.5–38.9) | <0.001* |

Age is shown as median age (the youngest age to the oldest age), and the data of Cr, sLC(I/U), $\beta_2$-MG, LDH, and Ig were shown as a median (25-75th percentiles).

*represents $P < 0.05$, with statistical difference.

ratio in the MM arm was significantly higher than that in the non-MM arm (P<0.001) (Table 1).

## The screening value of sLC ratio in the MM arm

The screening value of sLC ratio in the MM arm was analyzed by the ROC curve, and the AUC value was 0.875 (*P*< 0.0001) (Fig 1). The optimal screening cutoff value of sLC ratio was 3.2121, with a sensitivity of 81.16% and a specificity of 94.87% for the diagnosis of MM. The above-mentioned results suggest that sLC ratio has a high specificity for the diagnosis of MM, accompanying by a low sensitivity. It is necessary to combine sLC ratio with other clinical indicators to improve its sensitivity in MM screening.

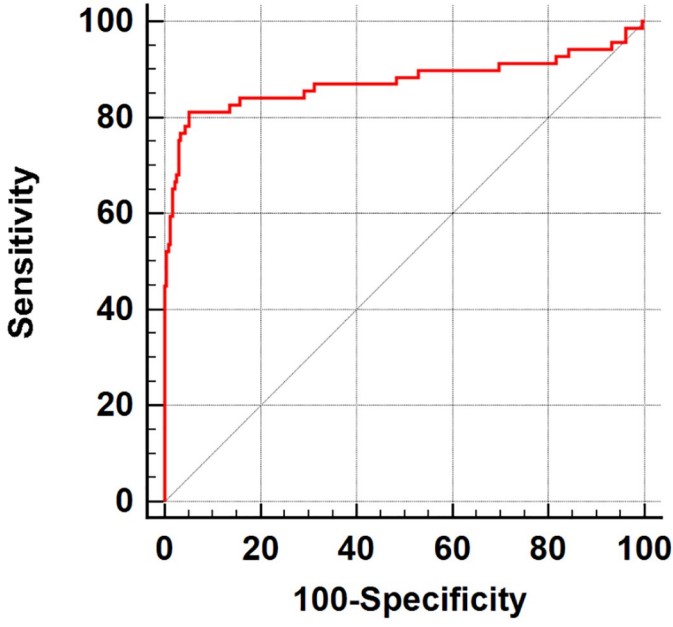

**Fig 1. ROC curve of sLC ratio.**

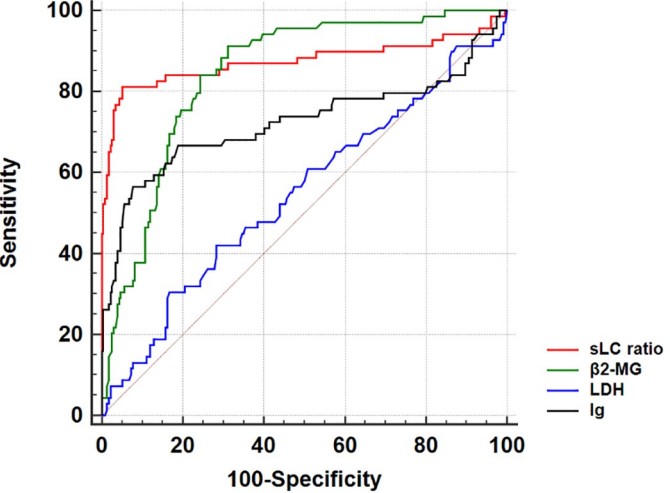

**Fig 2. ROC curve of sLC ratio, β₂-MG, LDH and Ig.**

## Other clinical indicators of MM

Comparison of the two arms showed that the levels of $\beta_2$-MG and Ig in the MM arm were significantly higher than those in the non-MM arm ($P<0.001$). However, there was no significant difference in LDH level between the two arms (P>0.05) (Table 1).

## The screening value of clinical indicators of MM

The screening value of $\beta_2$-MG, LDH, and Ig in the MM arm was analyzed by the ROC curve (Fig 2), and the AUC values were 0.843, 0.547, and 0.723, respectively. The optimal screening cutoff values were 1.95 mg/L, 220 U/L, and 46.4 g/L, respectively (Table 2). It was revealed that the screening accuracy of $\beta_2$-MG and Ig was high, and the difference was statistically significant (P<0.0001), while the sensitivity and specificity were still low.

## The screening value of combined MM-related clinical indicators

The screening value of combined MM-related clinical indicators was analyzed by the ROC curve (Table 3 and Fig 3). The triple combination of sLC ratio (3.2121), $\beta_2$-MG (1.95 mg/L), and Ig (46.4 g/L) yielded a higher screening value (AUC, 0.952) compared with that of sLC ratio alone or other combined indicators ($P<0.0001$). In the triple combination, the sensitivity and specificity were 94.20% and 86.75%, respectively, indicating that it had a high screening value for MM. The addition of LDH to the triple combination and formation of quadruple combination did not optimize the screening value, with AUC, sensitivity, and specificity of

**Table 2. Area under the ROC curve of sLC ratio, β2-MG, LDH and Ig.**

| Clinical indicators | AUC | Standard error | 95% confidence interval (AUC) | Optimal cut-off value | Sensitivity (%) | Specificity (%) | P |
|---|---|---|---|---|---|---|---|
| sLC ratio | 0.875 | 0.0337 | 0.832–0.910 | >3.2121 | 81.16 | 94.87 | <0.0001* |
| β₂-MG | 0.843 | 0.0248 | 0.797–0.882 | >1.95mg/L | 91.3 | 68.8 | <0.0001* |
| LDH | 0.547 | 0.0416 | 0.489–0.604 | >220U/L | 42.03 | 71.79 | 0.2627 |
| Ig | 0.723 | 0.0442 | 0.669–0.773 | >46.4g/L | 56.52 | 92.31 | <0.0001* |

* represents $P<0.05$, with statistical difference.

**Table 3. The area under the ROC curve analysis of the sLC ratio single indicator and combined with other indicators.**

| Clinical indicators | AUC | Standard error | 95% confidence interval (AUC) | Sensitivity (%) | Specificity (%) | $P_1$ | $P_2$ |
|---|---|---|---|---|---|---|---|
| sLC ratio | 0.875 | 0.034 | 0.832~0.910 | 81.16 | 84.87 | <0.0001* | |
| sLC ratio+β₂-MG | 0.949 | 0.016 | 0.918~0.871 | 92.75 | 86.75 | <0.0001* | 0.0054* |
| sLC ratio+LDH | 0.889 | 0.030 | 0.848~0.922 | 81.16 | 93.59 | <0.0001* | 0.2978 |
| sLC ratio+Ig | 0.870 | 0.035 | 0.827~0.906 | 81.16 | 94.44 | <0.0001* | 0.2305 |
| sLC ratio+β₂-MG+LDH | 0.950 | 0.161 | 0.918~0.971 | 94.20 | 85.47 | <0.0001* | 0.0056* |
| sLC ratio+β₂-MG+Ig | 0.952 | 0.015 | 0.922~0.973 | 94.20 | 86.75 | <0.0001* | 0.004* |
| sLC ratio+LDH+Ig | 0.887 | 0.031 | 0.846~0.920 | 81.16 | 93.59 | <0.0001* | 0.3646 |
| sLC ratio+β₂-MG+LDH+Ig | 0.952 | 0.015 | 0.922~0.973 | 94.20 | 85.47 | <0.0001* | 0.0043* |

* represents $P<0.05$, the difference is statistically significant. $P_1$ is the statistical significance of the area under the ROC curve of each indicator. $P_2$ is the statistical significance of the difference between sLC ratio and combined indicators in the context of the area under the ROC curve.

0.952, 94.20%, and 85.47%, respectively. Collectively, the triple combination is the best combination model for a simple, economic and efficient screening of MM.

## Discussion

It is well-known that MM is a malignant disease with the abnormal proliferation of clonal plasma cells. It is a common malignant tumor of the hematological system [1, 2]. It mainly occurs in the elderly population, it is still incurable, and its incidence has recently increased in China. Clonal plasma cells proliferate in the bone marrow, resulting in multiple osteolytic lesions, hypercalcemia, anemia, and kidney impairment. The onset of MM is relatively insidious, which can easily lead to missed diagnosis and delayed optimal treatment. Therefore, early screening is extremely important. At present, IFE, serum protein electrophoresis, and serum-free light chain (SFLC) are the main assays for diagnosing MM. However, these assays have not been widely utilized in Chinese hospitals, especially in county or district hospitals. As these assays are time-consuming and relatively expensive, they are not appropriate for MM screening.

Normal human serum immunoglobulins are composed of a pair of light chains and heavy chains. Clonal plasma cells secrete a single-type of immunoglobulin, resulting in an imbalanced light chain ratio. Consequently, an abnormal sLC ratio may suggest monoclonal proliferation of plasma cells [4, 5]. The SFLC assay is a sensitive antibody-based method for the detection of low concentrations of monoclonal-free light chains in serum. The normal FLC ratio ranges from 0.26 to 1.65, while approximately 90% of MM patients have abnormal FLC ratios [6, 7]. For MM patients without other symptoms, if the ratio of involved/uninvolved FLC is ≥100, the reported risk of progression with end-organ damage in the next two years is as high as 70–80% [8–10]. In these patients, the risk of progression within the next two years increases to 93% if the absolute value of the involved FLC is higher than 100 mg/dL. Given a high rate of progression, an FLC ratio of ≥100 is currently considered to have a screening value for MM [3]. However, FLC testing is not highly popular in Chinese hospitals, and it is not appropriate for MM screening due to its time-consuming and being expensive features. However, the sLC detection requires a shorter time and a lower cost, thus, it can be broadly performed in Chinese hospitals. Therefore, several MM-related clinical indicators including sLC were combined and analyzed to find out a simple, economic and efficient screening method for MM.

Normally, immunoglobulin is an immunologically active polyclonal antibody secreted by plasma cells. Infections, autoimmune diseases, hepatocyte diseases, etc. could increase the

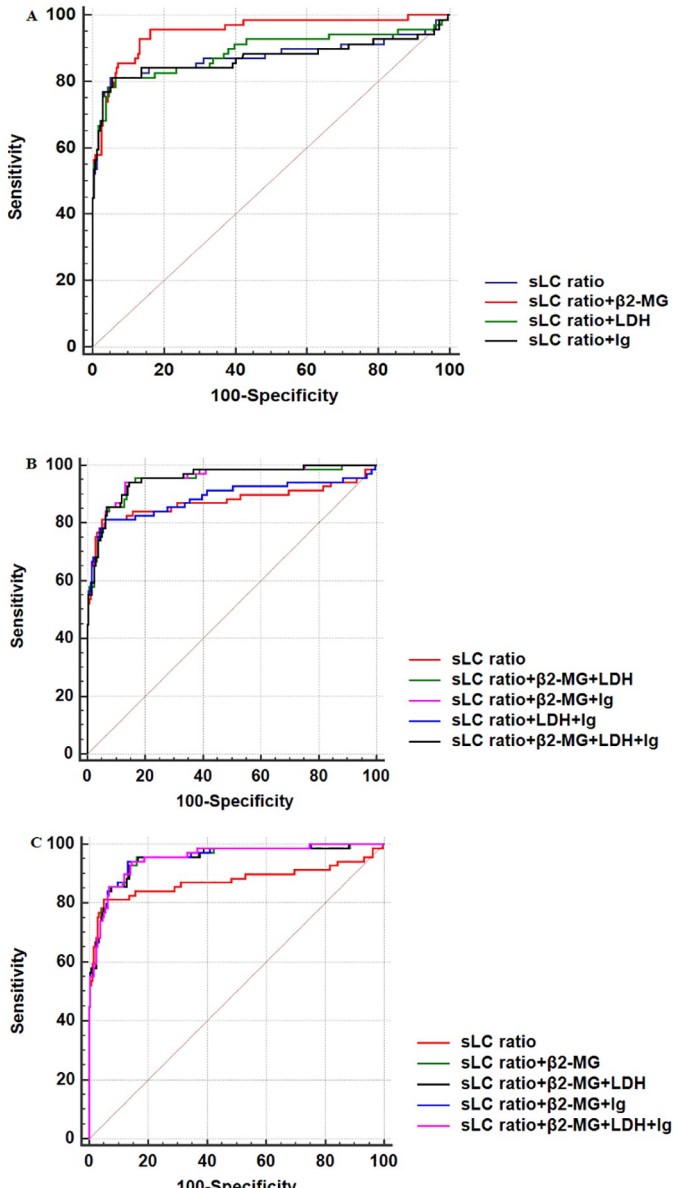

**Fig 3. ROC curve of different combined clinical indicators.** A, ROC curve of sLC ratio and two combined clinical indicators including sLC ratio.B, ROC curve of sLC ratio and three or four combined clinical indicators including sLC ratio. C, ROC curve of sLC ratio and other combined clinical indicators with an AUC>0.9.

value of immunoglobulin in a polyclonal manner. Although the absolute value of sLC varies, the sLC ratio remains normal. In the present study, the sLC ratio was used for MM screening, and the screening value of sLC ratio in MM was analyzed by the ROC curve. The AUC of sLC ratio was 0.875 ($P<$0.0001), the sensitivity was 81.16%, and the specificity was 94.87% indicating a high screening value for MM, while the sensitivity still needs to be improved. Therefore, different MM-related clinical indicators were combined and analyzed using the ROC curve. Ultimately, the triple combination (a combination of sLC ratio (3.2121), β₂-MG (1.95 mg/L), and Ig (46.4 g/L)) yielded a higher screening value, in which AUC was 0.952 and the sensitivity was as high as 94.20%, indicating that the mentioned combination model could meet the

demand of MM screening. This is a retrospective study, and further prospective studies with larger sample sizes should be therefore conducted.

In conclusion, the sLC ratio has a high screening value for MM with promising sensitivity and specificity, and it is of great significance for MM screening. The hybrid detection of sLC ratio, $\beta_2$-MG, and Ig can significantly improve the screening value and sensitivity, and increase the positive rate of MM. The combination of sLC ratio, $\beta_2$-MG, and Ig is the best combination model for a simple, economic and efficient screening of MM, and the best cut-off values were 3.2121, 1.95 mg/L, and 46.4 g/L, respectively.

## Supporting information

**S1 Data.**
(XLS)

## Author Contributions

**Conceptualization:** Limei Ying, Linglong Xu.

**Data curation:** Xiaochang Zhang, Nina Lu.

**Funding acquisition:** Linglong Xu.

**Investigation:** Xiaochang Zhang, Nina Lu, Lei Zhao, Yanfang Nie, Guofen Wang, Sai Chen, Linglong Xu.

**Methodology:** Limei Ying.

**Supervision:** Linglong Xu.

**Writing – original draft:** Limei Ying.

**Writing – review & editing:** Limei Ying, Xiaochang Zhang, Nina Lu, Lei Zhao, Yanfang Nie, Guofen Wang, Sai Chen, Linglong Xu.

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
