## [Decision Letter · Decision Letter 0]

9 Nov 2022

PONE-D-22-24567Evaluating the screening value of serum light chain ratio, β2 microglobulin, lactic dehydrogenase and immunoglobulin in patients with multiple myeloma using ROC curvesPLOS ONE

Dear Dr. Xu Linglong,

Thank you for submitting your manuscript to PLOS ONE. After careful consideration, we feel that it has merit but does not fully meet PLOS ONE’s publication criteria as it currently stands. Therefore, we invite you to submit a revised version of the manuscript that addresses the points raised during the review process.

 We appreciate your study which is an interesting study. However, the manuscript does not meet the publication criteria as the study design and data should be appropriate and described in sufficient detail. Please carefully consider and respond all of the reviewer’s comments and suggestions.

We look forward to receiving your revised manuscript.

Kind regards,

Vipa Thanachartwet, M.D.

Academic Editor

PLOS ONE

Journal Requirements:

"The study was supported by the Scientific Project of Taizhou Science and Technology Bureau (No. 20ywa31)."

"Linglong Xu received a grant from Taizhou Science and Technology Bureau（No. 20ywa31）

6. Please include your tables as part of your main manuscript and remove the individual files. Please note that supplementary tables (should remain/ be uploaded) as separate "supporting information" files.

Reviewers' comments:

Reviewer's Responses to Questions

**Comments to the Author**

1. Is the manuscript technically sound, and do the data support the conclusions?

Reviewer #1: Partly

2. Has the statistical analysis been performed appropriately and rigorously? 

Reviewer #1: Yes

3. Have the authors made all data underlying the findings in their manuscript fully available?

Reviewer #1: Yes

4. Is the manuscript presented in an intelligible fashion and written in standard English?

Reviewer #1: No

5. Review Comments to the Author

Reviewer #1: This is a retrospective study of 303 suspected MM patients. The finding shows the triple combination (sLC

ratio+β2-MG+Ig) had a sensitivity of 94.20% and a specificity of 86.75%. I have some comments

1. Due to small number of cases, please provide smaple size caluculation to eveluate that the number of patients is optimal for this analysis

2. Patient in MM group shold be exclude patient with SFLC ratio >100 to compare with non-MM group because this study is objective to find factor predictive of MM dignosis. But patient with SFLC >100 can diagnosis of MM and do not need other lab parameters (β2-MG+Ig) to predict the diagnosis.

3. Beta-2 microgloulin might be false high in patients with renal impairment. Therefore please provide baseline creatinine and renal function between both groups.

6. PLOS authors have the option to publish the peer review history of their article (what does this mean?). If published, this will include your full peer review and any attached files.

Reviewer #1: No

---

## [Author Response · Author response to Decision Letter 0]

24 Dec 2022

We respond to the reviewer and the editor comments in the "response to reviewer" and the "cover letter".

---

## [Editor Report · Decision Letter 1]

31 Jan 2023

Evaluating the screening value of serum light chain ratio, β2 microglobulin, lactic dehydrogenase and immunoglobulin in patients with multiple myeloma using ROC curves

PONE-D-22-24567R1

Dear Dr. Xu Linglong,

We’re pleased to inform you that your manuscript has been judged scientifically suitable for publication and will be formally accepted for publication once it meets all outstanding technical requirements.

Kind regards,

Vipa Thanachartwet, M.D.

Academic Editor

PLOS ONE

Additional Editor Comments (optional):

All issues raised by the reviewer have been addressed.
---

## [Editor Report · Acceptance letter]

8 Feb 2023

PONE-D-22-24567R1 

Evaluating the screening value of serum light chain ratio, β_2_ microglobulin, lactic dehydrogenase and immunoglobulin in patients with multiple myeloma using ROC curves 

Dear Dr. Xu:

I'm pleased to inform you that your manuscript has been deemed suitable for publication in PLOS ONE. Congratulations! Your manuscript is now with our production department. 

Kind regards, 

on behalf of

Associate Professor Vipa Thanachartwet 

Academic Editor

PLOS ONE